# Hydrogenation of High-Density Polyethylene during Decompression of Pressurized Hydrogen at 90 MPa: A Molecular Perspective

**DOI:** 10.3390/polym15132880

**Published:** 2023-06-29

**Authors:** Mina Kim, Chang Hoon Lee

**Affiliations:** 1Grad. School of Chemical Engineering, Chosun University, Chosundae-5-gil, Dong-gu, Gwangju 61452, Republic of Korea; ma2011@naver.com; 2Department of Biochemical Engineering, Chosun University, Chosundae-5-gil, Dong-gu, Gwangju 61452, Republic of Korea

**Keywords:** hydrogenation, partial hexagonal phase, 90 MPa hydrogen pressure

## Abstract

To investigate changes in the physical and chemical properties of high-density polyethylene (HDPE) upon the rapid release of hydrogen gas at a pressure of 90 MPa, several characterization techniques have been employed, including optical microscopy, scanning electron microscopy, X-ray diffraction, differential scanning thermal analysis, and attenuated total reflectance Fourier-transform infrared spectroscopy. The results showed that both physical and chemical changes occurred in HDPE upon a rapid release of hydrogen gas. Physically, a partial hexagonal phase was formed within the amorphous region, and the overall crystallinity of HDPE decreased. Chemically, hydrogenation occurred, leading to the addition of hydrogen atoms to the polymer chains. Oxidation also occurred, for example, the formation of ester -C=O groups. Crosslinking and an increase in -CH_3_ end termination were also observed. These changes suggest that structural transformation and chemical modification of HDPE occurred upon the rapid release of hydrogen gas.

## 1. Introduction

Global warming, caused by the accumulation of greenhouse gases, is the most pressing problem facing mankind in the 21st century. Therefore, efforts to reduce greenhouse gas emissions are underway globally. CO_2_, which has become synonymous with greenhouse gases, has increased rapidly since the industrial revolution, with the advent of hydrocarbon-based internal combustion engines. Among the various approaches for reducing CO_2_ emissions, converting a hydrocarbon-based internal combustion engine into an electric vehicle (EV) or a fuel-cell electric vehicle (FCEV) is the most direct method. In a hydrogen fuel-cell vehicle, hydrogen must be stored in a gaseous state in a hydrogen pressure vessel capable of withstanding a pressure of 70 MPa owing to its high energy density. It uses a Type 4 tank padded with a high-density polyethylene (HDPE) liner because HDPE is lightweight and flexible and has excellent resistance to chemicals; furthermore, it can withstand high pressures and temperatures [1,2,3,4,5,6,7]. Here, the carbon fiber exterior does not come into direct contact with hydrogen and must maintain the mechanical strength required at a hydrogen pressure of 70 MPa. The HDPE liner is in direct contact with hydrogen, and thus should play a role in preventing the leakage of hydrogen owing to the pressure difference between the inside and outside of the hydrogen pressure vessel. Because hydrogen penetrates HDPE during hydrogen charging and is released again during hydrogen discharge, mechanical, physical, and chemical deteriorations may occur in HDPE owing to dissolved hydrogen.

According to previous studies available in the literature, polymeric materials have been observed to undergo morphological deformation during the cyclic processes of high-pressure hydrogen gas charging and discharging [7,8,9]. Initially, hydrogen blistering occurs in the form of a single spherical bubble, which further develops into multiple bubbles. Some of these bubbles return to their original morphology without any cavitation structure, whereas others ultimately form a permanent cavitation structure [7,8,9,10]. Furthermore, the rupture of the main chain has been evidenced with the introduction of a fluorescent moiety at specific crosslinking sites between the nearest neighboring chains. In the case of HDPE, the initiation and development of blisters as well as the formation of the final cavitation structure have been extensively investigated using digital cameras. However, there is still a dearth of studies examining the potential differences between HDPE with and without cavitation structures [11,12,13]. Therefore, it is necessary to gain a comprehensive understanding of HDPE at the molecular level, considering both the presence and absence of void structures.

This study aimed to investigate the physical and chemical potential differences between HDPE with and without void structures after a rapid release of hydrogen gas at 90 MPa. The properties of neat HDPE before the hydrogen treatment were compared with those of HDPE after the hydrogen treatment (henceforth referred to as H_2_-HDPE).

## 2. Materials and Methods

### 2.1. HDPE Preparation

The bead-type HDPE used in this experiment was commercially available (HDPE bead, LG Chem, Seoul, South Korea). The procured HDPE had a density of 0.955 g/cm^3^ and a melt flow index of 0.028 g/min.

### 2.2. Hydrogen Pressure Treatment

HDPE beads with diameter and thickness of 13 and 3 mm, respectively, were subjected to hydrogen gas at a pressure of 90 MPa for more than 90 min. Subsequently, the chamber valve was opened, leading to a rapid release of H_2_ within 60 s. The polymer infrastructure in hydrogen fuel-cell cars is typically used at 90 MPa in hydrogen refueling stations or 70 MPa in hydrogen pressure vessels. Therefore, in the first experiment, a maximum pressure of 90 MPa was selected.

### 2.3. Microscopy Analysis on Surface Morphology

The surface morphology was examined using three different microscopy techniques, namely optical microscopy (OM; ECLIPSE MA200, Nikon, Tokyo, Japan), polarized optical microscopy (POM; Jenalab^pol^, ZEISS, Jena, Germany), and scanning electron microscopy (SEM; S-4800, HITACHI, Japan).

### 2.4. X-ray Diffraction (XRD)

HDPE beads were used for XRD experiments. Each diffraction pattern was obtained within a 2θ range of 3 to 90°, using a Bragg–Brentano configuration (XPERT-PRO TT, Panalytical, Malvern, England) with Cu K-α radiation (λ = 1.5406 Å). The step size was set to 0.01° and each scan lasted 1 s.

### 2.5. Differential Scanning Calorimetry (DSC)

Differential scanning calorimetry (DSC) measurements were performed using Shimadzu equipment (DSC-60A, Shimadzu, Kyoto, Japan). Approximately 5 mg of each sample was enclosed in a standard aluminum pan and subjected to a temperature range of 30 to 300 °C, with a heating rate of 2 °C/min under a nitrogen gas atmosphere.

### 2.6. Fourier Transform Infrared Spectroscopy (FTIR)

Infrared spectroscopic analyses of both HDPE and H_2_-HDPE beads were conducted using a spectrometer (Thermo Fisher Scientific, Waltham, MA, USA). Knife-cut beads were scanned in the wavenumber range of 4000–400 cm^−1^ using the attenuated total reflection (ATR) mode. The analysis consisted of 32 scans, performed at a resolution of 4 cm^−1^.

## 3. Results and Discussion

Upon rapid decompression of the hydrogen pressure from 90 to 0.1 MPa, a noticeable change occurred in the appearance of every H_2_-HDPE bead. A white trajectory was left behind, with a colorless and transparent background, as depicted in Figure 1.

The position of the white trajectory within each H_2_-HDPE bead varied significantly. However, the trajectory typically originated from a relatively central region and extended towards the outer surface of the HDPE bead. For the experiments, the H_2_-HDPE sample was divided into three pieces, cut perpendicular to the longitudinal axis of the bead, as shown in Figure 1b. Piece #1 represents the portion that included the termination point of the white trajectory and it was used to examine the surface morphology at that specific location. Piece #2 was used for OM, wide-angle X-ray diffraction (WXRD), and attenuated total reflection Fourier-transform infrared (ATR-FTIR) spectroscopy. Piece #3 was designated for DSC analysis. This experimental arrangement is clearly illustrated in Figure 1b.

### 3.1. Microscopy of Surface Morphology

Initially, we selected Piece #1, which included the terminated surface of the white trajectory, and used OM to observe the surface morphology at the endpoint of the white trajectory. The surface morphology revealed weakly collapsed structures, indicating a pathway for the outburst of high-pressure hydrogen gas. SEM was performed for a more detailed analysis, and the results are shown in Figure 2. It is evident from Figure 2a that the entire fracture exhibited numerous tone-burst structures, indicating the occurrence of a polymer chain scission. A further magnification revealed that the orientation of the tone-burst structures was nearly parallel to the lamellar thickness, which, in turn, was aligned with the polymer chain direction, as shown in Figure 2b.

Figure 3a shows a digital camera image obtained from one cross-section of the central piece (Piece #2), which includes the evolution of the white trajectory. The white section in Figure 3b shows a cavity structure with two different spherical holes with radii of the order of μm formed in its inner space (henceforth referred to as H_2_-HDPE-W). In the outer part of the white cavity, which is henceforth referred to as H_2_-HDPE-T, the formation of channel structures during the release of the gas is evident, particularly noticeable in the lower-left corner of Figure 3c. Within the region indicated by the red line in Figure 3c, the polymer morphology exhibits a distinct semi-transparent arch-like bridge structure. This bridge structure is composed of a thin two-dimensional ribbon with embedded fibrils that maintain a consistent orientation. The bridge is surrounded by a transparent amorphous matrix [14,15]. This unique morphology extends to the left side of Figure 3c, where the bundles of fibers forming the ribbon structure are observed to be in a completely relaxed state and devoid of any elasticity. The presence of these structures indicates that the rapid decompression and subsequent gas release caused significant changes in the polymer morphology and a loss of elasticity. The formation of channels and an arch-like bridge structure suggest a complex rearrangement and deformation of the polymer during the gas release. Additionally, the splitting of the polymer morphology in the fully relaxed state into three branches, particularly in the lower-left corner, resembles the appearance of the polymer being torn apart owing to the rapid release of hydrogen pressure. This morphology indicates significant chain stretching and expansion of the polymer structure during the rapid decompression. The observed splitting and tearing suggest that the polymer experienced high mechanical stress and strain during the hydrogen release, thus leading to the deformation and rupture of the polymer chains.

### 3.2. XRD

To determine the crystal structure responsible for H_2_-HDPE-W, WXRD analysis was performed. Figure 3a illustrates the clear separation between the white and transparent regions, thus enabling independent WXRD data collection for each region. Figure 4 displays the WXRD patterns obtained from neat HDPE, and the white cavities of H_2_-HDPE-W and H_2_-HDPE-T. The crystal structure of HDPE at the ambient temperature and 1 atm is orthorhombic and belongs to the P_nam_ space group, with lattice constants of a = 7.6 Å, b = 4.3 Å, and c = 2.4 Å. Diffraction angles of 2θ = 21.4 and 23.8° correspond to the (110)_o_ and (200)_o_ planes, respectively [16,17]. The white cavities of H_2_-HDPE-W and H_2_-HDPE-T exhibit nearly identical major XRD patterns as that of neat HDPE, although the diffraction intensity is reduced for each Miller index. In the case of H_2_-HDPE-T, the peak positions of the (110)_o_ and (200)_o_ planes are shifted by approximately Δ2θ = 0.02° towards the higher angles, indicating a compressive lattice deformation caused by the high-pressure hydrogen. Another interesting observation is the presence of a small but distinct peak at 2θ = 20.8° in the XRD pattern of H_2_-HDPE-W. This peak corresponds to the diffraction angle associated with the hexagonal crystal lattice of (100)_h_ [14,18,19,20,21,22,23]. The presence of this peak suggests the partial existence of an extended chain crystal structure within the hexagonal phase of H_2_-HDPE-W.

Once the XRD results are deconvoluted with three Gaussian functions to obtain the respective areas and half-height widths, the % crystallinity can be calculated by substituting the obtained areas into Equation (1) given below [24].
(1)$$\%X_{c}=\frac{A_{cr}}{A_{cr}+A_{am}}\times 100\%(A_{cr}=A_{110}+A_{200}:Crystalline;\;A_{am}:Amorphous\;area)$$

Next, the average size L of the HDPE crystals perpendicular to the Miller index planes (110)_o_ and (200)_o_ from the full width at half maximum (FWHM) of the XRD line can be calculated using the half-height width and Scherrer equation (Equation (2)) given below [25]:(2)$$L=\frac{K\cdot \lambda }{\beta \cdot cos\theta }(\beta :Half\;width\;in\;radians;\;\lambda :\text{X-ray}\;wavelength;\;K=0.89;\;\theta =a\;diffraction\;angle)$$

The results are summarized in Table 1, which reveals that the crystallinity of the orthorhombic crystal structure decreased from 65% before the hydrogen pressure treatment to 57% for H_2_-HDPE-W and 47% for H_2_-HDPE-T after the hydrogen pressure treatment. But the lamellar thickness of the (110)_o_ plane increased from 187 Å before the hydrogen treatment to 248 Å for H_2_-HDPE-W and 263 Å for H_2_-HDPE-T. Similarly, the lamellar thickness of the (200)_o_ plane increased from 167 Å before the hydrogen treatment to 208 Å for H_2_-HDPE-W and 221 Å for H_2_-HDPE-T. In terms of the d-spacing value, only H_2_-HDPE-T showed a decrease of approximately 0.02 Å in both the (110)_o_ and (200)_o_ planes, indicating further compressive folding during the rapid decompression. On the other hand, the I(200)_o_/I(110)_o_ ratio, which represents the chain stretching ratio, was 0.31 for neat HDPE, 0.53 for H_2_-HDPE-W, and 0.37 for H_2_-HDPE-T [26]. This indicates that the white cavity region of H_2_-HDPE-W exhibited a higher degree of chain stretching than that of H_2_-HDPE-T. The increase in the I(200)_o_/I(110)_o_ ratio suggests that the hydrogen pressure treatment influenced the molecular arrangement and promoted the formation of an extended chain crystal structure in the white cavity region of H_2_-HDPE-W. The stretching ratio of 0.53 observed in H_2_-HDPE-W was significantly higher than that obtained from ultra high molecular weight polyethylene (UHMWPE) produced using the melt-drawn method at elevated temperatures. This indicates that the chain stretching in H_2_-HDPE-W was exceptionally pronounced. Moreover, the completely relaxed state image in Figure 3d serves as a visual representation of the internal structural changes in the polymer.

The WXRD study revealed that in the white cavity region of H_2_-HDPE-W, the folding degree of the polymer chain was similar to that of neat HDPE; however, the stretching ratio was remarkably high. In contrast, in the other transparent region of H_2_-HDPE-T, the molecular folding was even more compact than that in neat HDPE, while the stretching degree was slightly higher. This suggests that the crystalline lamellae in the white cavity of H_2_-HDPE-W were subjected to an extensive stretching force, whereas the lamellae in the other transparent regions of H_2_-HDPE-T experienced compression during the rapid decompression process.

The observed changes in the polymer morphology and crystal structure, as indicated via XRD and microscopy results, suggest significant modifications in the arrangement of the polymer chains. These modifications were likely influenced by the hydrogen pressure treatment and subsequent rapid decompression process. Therefore, the completely relaxed state observed in Figure 3d can be attributed to the rearrangement and stretching of the polymer chains, which might have led to the formation of a partially chain-extended hexagonal phase embedded in an amorphous background matrix.

### 3.3. DSC

The results of the DSC experiments on Piece #3, which included the central starting point and neat HDPE, are depicted in Figure 5. Following the high-pressure hydrogen treatment, the most notable change is the occurrence of melting behavior at two different temperatures. One is the prominent melting peak at 131 °C, attributable to the orthorhombic crystal structure, while the other is a new minor weak peak that appeared at approximately 165 °C (as shown in the inset of Figure 5).

For the main melting peak, the % crystallinity (%X_dsc_) can be calculated using the following equation:(3)$$\%X_{dsc}=\frac{{∆H}_{m}}{{∆H}_{m0}}\times 100$$
where, ΔH_m_ is the measured heat of fusion and is equal to the area under the DSC melting curve, and ΔH_m0_ is the unit mass heat of fusion of 100% crystalline HDPE (293 J/g) [16,27]. Based on Equation (3), the crystallinity before hydrogen pressure treatment was calculated as 77% (ΔH_m_ = 226.49 J/g), while the crystallinity after hydrogen pressure treatment decreased to 74% (ΔH_m_ = 217.04 J/g).

The lamellar thickness L_c_ of the crystal grains can be calculated using the Gibbs–Thomson equation [28].
(4)$$L_{c}=\frac{2\sigma_{e}T_{m0}}{∆H_{m0}(T_{m0}-T_{m})}$$
where, σ_e_ is the surface free energy (0.093 J/m^2^), ΔH_m0_ is the volumetric heat of fusion (2.88 × 108 J/m^3^), and Tm_0_ is the melting temperature of an infinitely thick HDPE crystal (414.6 K). $L_{c}$ was approximately 311 Å (132 °C) before the hydrogen pressure treatment, and it remained almost the same at 311 Å (131 °C) after the hydrogen pressure treatment. In the case of polyethylene (PE), a decrease in the crystallinity implies a reduction in the volume of the crystalline region where -(CH_2_)- chains fold in parallel. This reduction leads to a decrease in the number of molecules participating in the van der Waals interactions between the folded -(CH_2_)- chains in the crystalline region, which should proportionally lower the melting temperature. However, in our results, although the hydrogen pressure treatment reduced the orthorhombic crystallinity by approximately 3%, the change in the melting temperature was negligible within the experimental error (±0.5 °C).

Next, we considered the minor weak peak shown in the inset of Figure 5. This peak, observed at 165 °C, cannot be attributed to the orthorhombic structure, as the fully orthorhombic crystal structure of HDPE typically exhibits a melting temperature of at least 414.6 K (141.6 °C) and a heat of fusion of 293 J/g. The enthalpy of fusion corresponding to the integral area was 6.07 J/g. The melting temperature of HDPE (165 °C) closely matched that observed in the case of the hexagonal crystal structure [14,29,30]. This observation agreed with the experimental findings from OM and WXRD analyses, which indicated the presence of a hexagonal phase in the treated HDPE sample. The similarity in the melting temperatures further supported the inference that the hydrogen pressure treatment and subsequent structural modifications influenced the crystalline arrangement of the polymer, leading to the formation of a hexagonal crystal structure with altered thermal properties [31].

The DSC analysis showed that HDPE exhibited two distinct melting phenomena. The first could be attributed to the ambient orthorhombic crystal structure with a melting temperature of 131 °C, while the second corresponded to a partial hexagonal crystal structure with a melting temperature of 165 °C. The % crystallinity associated with the low-temperature melting of the ambient orthorhombic crystal structure showed a slight decrease, indicating some structural modifications past the hydrogen pressure treatment. However, the crystalline lamellar thickness remained unchanged within the experimental error. Further details and summarized results are presented in Table 2.

According to the P–T phase diagram [17,32,33,34], pure HDPE can exist in three different physical states, namely the orthorhombic crystal structure, which is the most common state under the ambient pressure (1 atm); hexagonal crystal structure, which is typically observed under a high-pressure (>GPa) and high-temperature molten extended chain conditions, and liquid state [35]. Additionally, a monoclinic crystal structure is also possible, which can occur as a metastable phase under highly stressed ambient conditions or as a hysteretic phase at high pressures and temperatures [36,37,38]. In our case, the aforementioned conditions were not met. However, it appears that the presence of the hexagonal phase was observed. Therefore, it is necessary to investigate whether the hexagonal phase is possible under conditions other than those specified in the P–T phase diagrams. Previous research has shown that introducing crosslinking through radiation exposure can significantly lower the temperature required for hexagonal-phase formation [39]. Additionally, it has been reported that upon embedding orthorhombic-phase HDPE in an epoxy matrix [15], a transition to the hexagonal phase occurred at temperatures lower than those indicated in the P–T phase diagram. However, it is important to note that in both the cases investigated in this study, the temperature required to exhibit the hexagonal phase was considerably higher than the ambient temperature. Therefore, the currently observed hexagonal phase could not be explained by the existing research findings, thus warranting further investigation.

### 3.4. ATR-FTIR

The chemical changes at the molecular level resulting from the hydrogen pressure treatment were verified for Piece #2 using ATR-FTIR spectroscopy, and are depicted in Figure 6. In simple terms, the wavenumbers of the signal are shown in the figure, and the corresponding assignments for the signals are summarized in Table 3. The ATR-FTIR spectrum of neat HDPE primarily identified the signals in four distinct regions. First, the signals at 2915 and 2848 cm^−1^ correspond to the asymmetric and symmetric stretching vibrations of the methylene (–CH_2_–) groups, respectively. The signals at 1462 and 1472 cm^−1^ represent the bending deformation of the methylene groups [40,41].

The presence of wagging vibrations of CH_2_ in the crystalline region is indicated by the signal at 1367 cm^−1^. Additionally, the rocking vibration of the methylene group is observed in the 700–800 cm^−1^ range [41,44,45]. The broad signals at 3393 and 1590 cm^−1^ are attributed to the symmetric stretching and bending modes of the OH groups involved in the hydrogen bond network, respectively [50]. Furthermore, the small but distinct peaks can be assigned as follows.

The peaks at 1169 and 1261 cm^−1^ correspond to the symmetric and asymmetric bending deformations of the -CCH-groups, respectively [44]. These peaks suggest the presence of a small amount of crosslinking in neat HDPE. The peak at 1367 cm^−1^ corresponds to the wagging motion of the -CH_3_ groups, thus confirming their presence [41]. The two distinct peaks at 1045 and 1119 cm^−1^ can be attributed to the secondary and tertiary alcohol groups, respectively [43]. These assignments hold true for both the H_2_-HDPE-W and H_2_-HDPE-T samples.

Additionally, in the hydrogen-treated samples, along with the signals that define neat HDPE, there were newly emerged signals as well as signals that became more pronounced. These signals indicate the presence of additional functional groups or structural changes in the polymer. Notably, the FTIR signals observed for both the H_2_-HDPE-W and H_2_-HDPE-T samples yielded similar patterns, suggesting that the hydrogen treatment induced similar chemical modifications in both the samples. Among the new peaks, the signals at 3391 and 2019 cm^−1^ can be assigned to the R-C≡C-H groups, indicating the presence of alkyne groups. The pairs of signals at 3187, 1645, and 875 cm^−1^ can be attributed to the R-CH=CH_2_ groups, indicating the presence of vinyl groups [47,49]. The signals at 1377, 1302, and 804 cm^−1^ correspond to the -CH_3_ groups in the R-CH=CH-CH_3_ moiety of HDPE [41,43,51]. These spectral changes can be explained by the introduction of hydrogenation reactions involving radical species generated after the chain rupture during the rapid decompression process. In other words, the signal at 1741 cm^−1^ can be attributed to the ester -C=O groups, which might have resulted from oxidation [48,50]. However, the specific assignment of the 1420 cm^−1^ signal has not been reported till date. To gain a better understanding, it is important to revisit the broad FTIR signal observed at 1429 cm^−1^ in neat HDPE. This signal is typically associated with the scissoring vibration of the -CH_2_–groups in the amorphous region of HDPE, which consists of disordered and randomly oriented polymer chains. When the polymer undergoes rapid decompression from high-pressure hydrogen treatment, the chains experience reorientation, resulting in a change in the degree of order within the amorphous region. This reorientation can cause a narrowing and a shift of the FTIR signal, resulting in the observed signal at 1420 cm^−1^. In some cases, this signal has also been observed as a peak at 1430 cm^−1^ [33] during the phase transition from the orthorhombic structure to the monoclinic structure, or at 1450 cm^−1^ [33,52] in the case of the extended chain hexagonal structure. For our specific case, considering the experimental results discussed earlier, it is more reasonable to attribute the observed signal at 1420 cm^−1^ to the formation of the hexagonal phase in the amorphous region. This suggests that the reorientation and ordering of the polymer chains during the decompression of high-pressure hydrogen can potentially result in the formation of a hexagonal structure within the amorphous phase; this in turn, implies a significant change in the molecular arrangement and packing of the polymer chains, which can be detected by the vibrational modes associated with the scissoring vibrations of the -CH_2_- groups.

In the case of signals that have become more pronounced, for example, the two prominent peaks at 1169 and 1261 cm^−1^, these can be attributed to the crosslinking effects [44]. These signals indicate the symmetric and asymmetric bending deformations of the -CCH- groups, which were caused by an increase in the number of crosslinks in the polymer structure. Additionally, the signal at 1367 cm^−1^ corresponds to the wagging motion of the -CH_3_ groups, indicating an increase in the number of -CH_3_ end groups in the polymer chains [41]. Therefore, the enhanced signals at 1169, 1261, and 1367 cm^−1^ suggest that hydrogen treatment led to an increase in the number of crosslinks and -CH_3_ end groups in the polymer.

Subsequent to the high-pressure hydrogen treatment, various chemical changes occurred in H_2_-HDPE, including the introduction of new functional groups and an increase in the crosslinking and -CH_3_ end groups. These modifications are summarized in Table 4. The aforementioned treatment led to the formation of additional functional groups, such as R-C≡C-H and R-CH=CH_2_, as indicated by the appearance of new peaks in the FTIR spectrum. Moreover, the presence of enhanced peaks at 1169 and 1261 cm^−1^ suggests an increase in the crosslinking effects, whereas the intensified signal at 1367 cm^−1^ indicates an increase in the -CH_3_ end groups [53].

The change in the crystallinity of the HDPE samples owing to the hydrogen pressure treatment can be calculated using Equation (5) together with the absorption intensities in the range of 1472 to 1462 cm^−1^ [54].
(5)$$\%X_{IR}=\frac{I_{\beta }-\frac{I_{\alpha }}{1.233}}{I_{\beta }+I_{a}}\times 100$$

Here, I_α_ and I_β_ are the signal intensities of the doublet shape at wave numbers 1472 and 1462 cm^−1^, respectively, and the ratio of these signal intensities for a sample with 100% crystallinity is known to be 1.233. The empirical rule presented by Zerbi et al. [54] provides an equation (Equation (5) given above) to calculate the amorphous content (% amorphous) based on the crystallinity [17,54]. Therefore, the crystallinity can be obtained as a complement of the amorphous content (1 − % amorphous). After analyzing the results, it was observed that the crystallinity decreased from 69% before the hydrogen pressure treatment to 63% after the treatment. This indicates a reduction in the degree of crystallinity of the HDPE sample owing to hydrogen treatment.

The rapid decompression of high-pressure hydrogen gas in the presence of HDPE can indeed significantly affect the polymer structure. When the gas pressure was suddenly released, the spherulite structure of the polymer underwent an explosive expansion. This expansion could result in the disruption of the polymer chains and formation of numerous cavities within the polymer matrix. The violent release of the gas pressure during the decompression could lead to the generation of high shear forces and mechanical stresses within the polymer. These forces could cause a chain scission, chain entanglement, and rearrangement of the polymer segments. Consequently, the polymer structure would be disrupted and cavities or voids would be formed within the material. The formation of these cavities could lead to a decrease in the density and an increase in the porosity of the material, thus affecting the mechanical properties, for example, causing a reduction in the stiffness and strength of the polymer. Furthermore, the cavities could act as stress concentrators, thereby potentially leading to localized deformation or failure under external loads.

## 4. Conclusions

The experimental results and discussions presented herein provide insights into the changes observed in high-density polyethylene (HDPE) when subjected to rapid decompression of high-pressure hydrogen gas.

From a physical standpoint, the formation of a partial hexagonal phase within the amorphous region of HDPE was observed, accompanied by a decrease in orthorhombic crystallinity. This suggests a reorientation and rearrangement of the polymer chains during the rapid release of hydrogen pressure. The expansion of the spherulite structure, disruption of the polymer chains, and the presence of cavity structures further aid these physical changes.

Chemically, hydrogenation, oxidation, crosslinking, and termination of the -CH_3_ end groups were identified as the primary chemical transformations induced by the hydrogen treatment. Hydrogenation resulted in the addition of hydrogen atoms to the polymer chains, whereas oxidation led to the formation of oxidized groups, including ester -C=O groups. Crosslinking was evident from the characteristic bending deformations of the -CCH- groups; furthermore, an increase in the number of -CH_3_ end groups was observed through the wagging motion of the -CH_3_ groups.

Previous studies have only observed the occurrence of cavitation in HDPE after a high-pressure hydrogen treatment. However, this study demonstrated that, in addition to cavitation, there were concurrent alterations in the physical crystalline structure of HDPE and other associated chemical transformations after the chain scission. These findings offer molecular-level insights that can be utilized in the design and comprehension of the durability of pressure vessels in hydrogen fuel-cell vehicles. Further studies are warranted to explore the detailed mechanisms underlying these changes, and their implications for the properties and performance of HDPE materials.

## Figures and Tables

**Figure 1 polymers-15-02880-f001:**
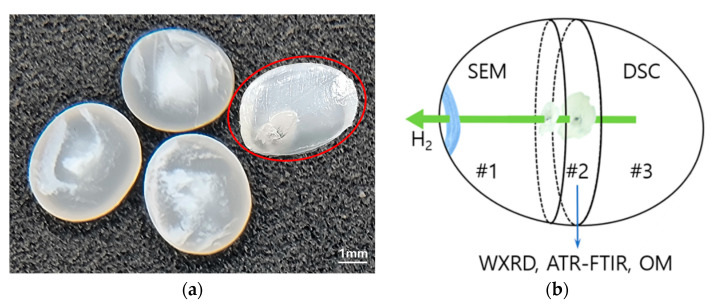
(**a**) White trajectory formed inside H_2_-HDPE after hydrogen pressure treatment at 90 MPa; (**b**) schematic illustration showing the division of the sample within the red circle in (**a**) into three pieces. These were labeled as #1, #2, and #3 for conducting the experiments.

**Figure 2 polymers-15-02880-f002:**
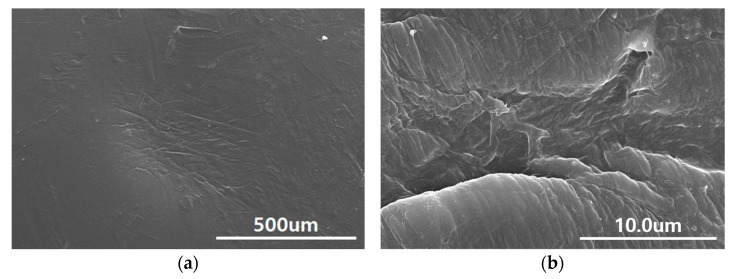
(**a**) SEM image of tone-burst structures observed on the surface of H_2_-HDPE after hydrogen treatment; (**b**) Image of an enlarged observation of one representative tone-burst structure.

**Figure 3 polymers-15-02880-f003:**
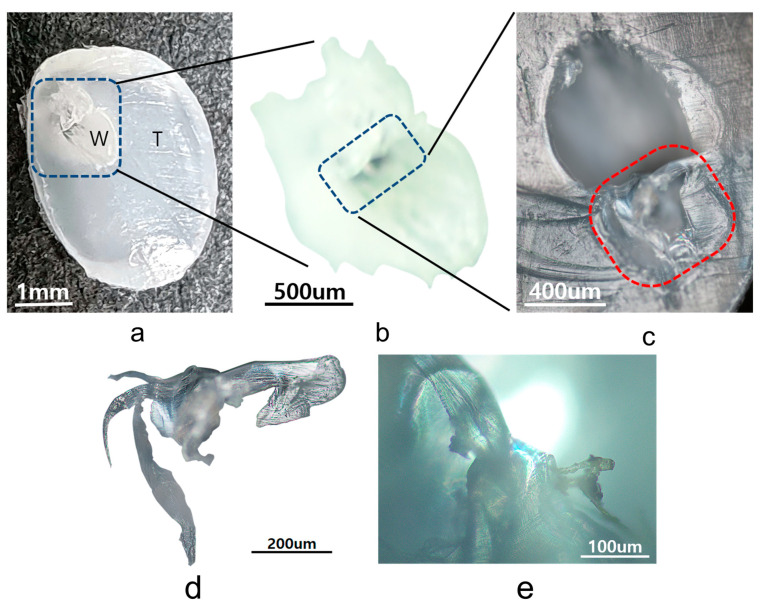
Images captured using a digital camera and OM to document changes following hydrogen pressure treatment: (**a**) Image of central sample (Piece #2). The white cavity region is outlined by blue dashed lines and labeled as “W,” while the other regions are labeled as “T”; (**b**) Zoomed image zoomed of area enclosed by blue dashed lines in (**a**), emphasizing the presence of two distinct holes within the white cavity. These holes are indicated by blue dashed lines; (**c**) Magnified image of the black-colored hole area indicated by the blue dashed lines in (**b**), obtained using OM; (**d**) Further-magnified image of the region indicated by the red dashed lines in (**c**); (**e**) POM image of the same area as in (**d**).

**Figure 4 polymers-15-02880-f004:**
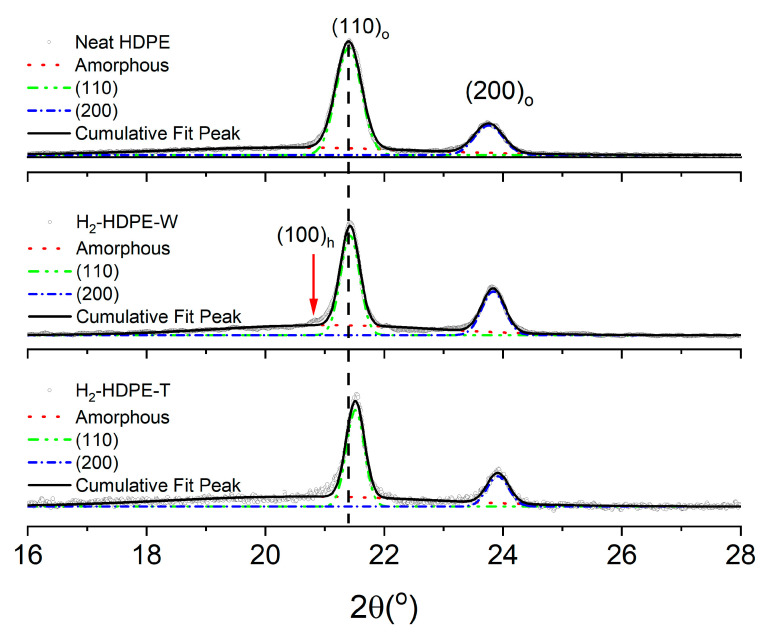
Results of WXRD for various regions of H_2_-HDPE, including the white cavity region of H_2_-HDPE-W and other areas of H_2_-HDPE-T, after undergoing hydrogen pressure treatment. WXRD results for neat HDPE prior to hydrogen treatment are also included for comparison. In the figure, (110)_o_ and (200)_o_ represent the (110) and (200) planes in the orthorhombic crystal structure, while (100)_h_ indicates the (100) plane in the hexagonal crystal structure.

**Figure 5 polymers-15-02880-f005:**
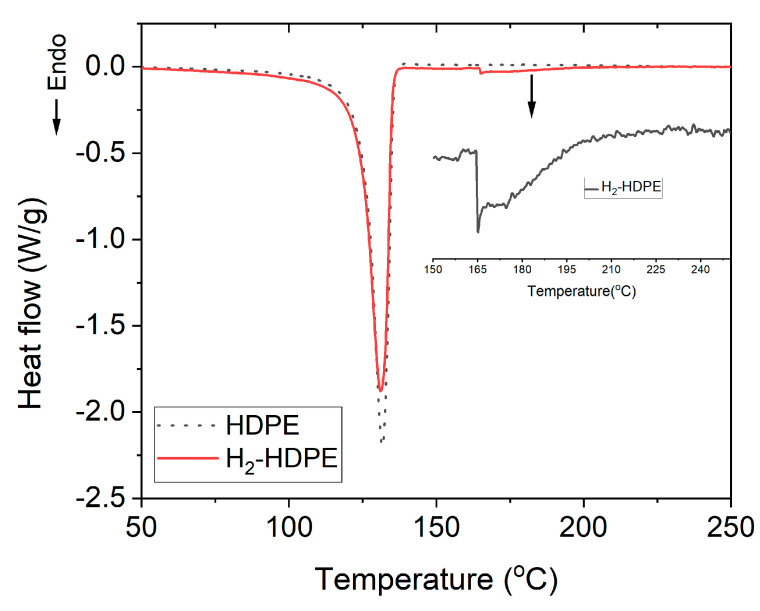
DSC results before and after hydrogen pressure treatment. Inset represents an enlarged view of melting transition at 165 °C.

**Figure 6 polymers-15-02880-f006:**
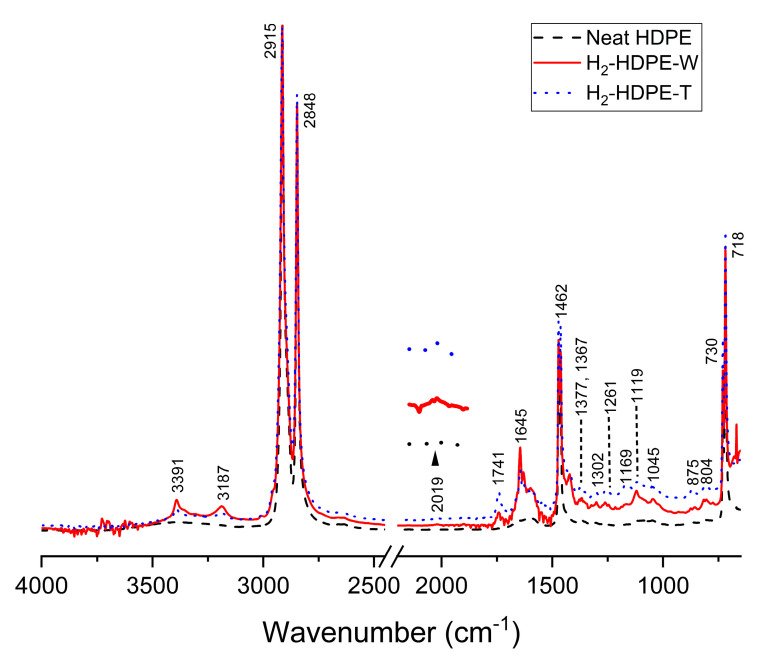
ATR-FTIR results obtained from neat HDPE, H_2_-HDPE-W, and H_2_-HDPE-T. Arabic numerals next to each signal indicate the corresponding wavenumbers of the signals. Inset figure, indicated by the upward arrow: enlarged view of the 2019 cm^−1^ region.

**Table 1 polymers-15-02880-t001:** XRD results for neat HDPE, H_2_-HDPE-W, and H_2_-HDPE-T.

XRD	%X_c_	Lamella Thickness (Å)		d-Spacing (Å)		I(200)/I(110)
		L_110_	L_200_	(110)	(200)	
Neat HDPE	65	187	167	4.15	3.74	0.31
H_2_-HDPE-W	57	248	208	4.13	3.72	0.53
H_2_-HDPE-T	47	263	221	4.15	3.73	0.37

**Table 2 polymers-15-02880-t002:** DSC results of HDPE and H_2_-HDPE.

DSC	Hm (J/g)	%X_c_	Melting Temperature (°C)	Thickness of Lamellae (Å)
Neat HDPE	226.49	77	132	311
H_2_-HDPE	217.04	74	131	311

**Table 3 polymers-15-02880-t003:** Functional groups assigned to wavenumbers of FTIR.

Assigned Wavenumber (cm^−1^)	Functional Group and Vibration Mode	New/Existing	Reference
718, 730	Split -CH_2_- (rocking)	Existing	[41]
804	-CH_3_ (bending)	New	[41]
875	-CH=CH_2_ (C-H bending)	New	[42]
1045	-CHOH- (C-O stretching)	Existing	[43]
1119	-COH(CH_3_)- (C-O stretching)	Existing	[43]
1169	Symmetric CCH bending	Existing	[44]
1261	Asymmetric CCH bending	Existing	[44]
1302	-CH_3_ (twisting and wagging)	New	[41]
1367	-CH_3_ (wagging)	Existing	[41]
1377	-CH_3_ (Symmetric C-H bending)	New	[42]
1462, 1472	-CH_2_- (bending)	Existing	[45]
1605	O-H (bending)	Existing	[46]
1645	C=C (stretching)	New	[47]
1741	C=O (ester)	New	[48]
2019	C≡C (stretching)	New	[49]
2848	-CH_2_- (Symmetric C-H stretching)	Existing	[41]
2915	-CH_2_- (Asymmetric C-H stretching)	Existing	[41]
3187	C=C-H (C-H stretching)	New	[47]
3351	O-H (stretching)	Existing	[46]
3391	C≡C-H (C-H stretching)	New	[49]

**Table 4 polymers-15-02880-t004:** Major chemical changes induced by high-pressure hydrogen treatment.

Chemical change	Molecular	Wavenumber (cm^−1^)
Hydrogenation	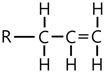	875, 1645, 3187
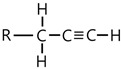	2019, 3391
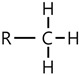	804, 1302, 1367, 1377
Crosslinking	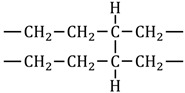	1169, 1261
Oxidation	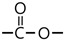	1741

## Data Availability

The data related to this study are available upon reasonable request.

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
