# Peer review of "Hydrogenation of High-Density Polyethylene during Decompression of Pressurized Hydrogen at 90 MPa: A Molecular Perspective"

_polymers, 2023, doi:10.3390/polym15132880_

Round 1

Reviewer 1 Report

This manuscript focused on investigating the changes in the physical and chemical properties of high-density polyethylene (HDPE) upon rapid release of hydrogen gas at a pressure of 90 MPa. This report is very interesting and can be published in this journal. But before publishing, some minor revisions are needed. Here are some detailed comments/questions for the authors:

1. In part 2.2 Hydrogen pressure treatment of the manuscript, why does the author choose such test pressure and test time? Can the author give an appropriate explanation?

2. It is recommended to add rulers to Figures 1a and 3.

3. It is suggested that the scales of SEM images be redrawn.

Reviewer 2 Report

This study aims to investigate the potential differences, both physical and chemical, between HDPE with and without void structures after the rapid release of hydrogen gas at 90 MPa. It will compare the properties of neat HDPE before hydrogen treatment with those of H2-HDPE after hydrogen treatment. To some extent, this is a topic of interest to the researchers in the related areas and this paper can be published after some minor revisions. The following points require consideration:

(1) The abstract is too long. Reformulate the abstract in order to clearly show the strengths of this work.

(2) Some sentences need reconstruction and the level of English should be improved.

(3) Please highlight the advantages and disadvantages of the findings.
